# Encapsulation of PI3K Inhibitor LY294002 within Polymer Nanoparticles Using Ion Pairing Flash Nanoprecipitation

**DOI:** 10.3390/pharmaceutics15041157

**Published:** 2023-04-06

**Authors:** Austin D. Fergusson, Rui Zhang, Judy S. Riffle, Richey M. Davis

**Affiliations:** 1Graduate Program in Translational Biology, Medicine, and Health, Virginia Tech, Blacksburg, VA 24061, USA; 2Department of Chemistry, Virginia Tech, Blacksburg, VA 24061, USA; 3Macromolecules Innovation Institute, Virginia Tech, Blacksburg, VA 24061, USA; 4Department of Chemical Engineering, Virginia Tech, Blacksburg, VA 24061, USA

**Keywords:** polymer nanoparticles, LY294002, flash nanoprecipitation, drug release

## Abstract

Flash nanoprecipitation (FNP) is a turbulent mixing process capable of reproducibly producing polymer nanoparticles loaded with active pharmaceutical ingredients (APIs). The nanoparticles produced with this method consist of a hydrophobic core surrounded by a hydrophilic corona. FNP produces nanoparticles with very high loading levels of nonionic hydrophobic APIs. However, hydrophobic compounds with ionizable groups are not as efficiently incorporated. To overcome this, ion pairing agents (IPs) can be incorporated into the FNP formulation to produce highly hydrophobic drug salts that efficiently precipitate during mixing. We demonstrate the encapsulation of the PI3K inhibitor, LY294002, within poly(ethylene glycol)-b-poly(D,L lactic acid) nanoparticles. We investigated how incorporating two hydrophobic IPs (palmitic acid (PA) and hexadecylphosphonic acid (HDPA)) during the FNP process affected the LY294002 loading and size of the resulting nanoparticles. The effect of organic solvent choice on the synthesis process was also examined. While the presence of either hydrophobic IP effectively increased the encapsulation of LY294002 during FNP, HDPA resulted in well-defined colloidally stable particles, while the PA resulted in ill-defined aggregates. The incorporation of hydrophobic IPs with FNP opens the door for the intravenous administration of APIs that were previously deemed unusable due to their hydrophobic nature.

## 1. Introduction

While the drug delivery field has evolved over the years by incorporating new technologies, successful delivery of hydrophobic therapeutics remains a difficult problem to overcome. Approximately 40% of all approved therapeutics are poorly soluble or completely insoluble in water, which limits bioavailability following administration [1]. Polymeric nanoparticles represent a promising solution to that problem. By encapsulating hydrophobic therapeutics within polymeric nanoparticles, it is possible to achieve much higher dosages of those therapeutics using various delivery methods—including intravenous and oral—than when the free therapeutic is administered. These nanocarriers also can be functionalized for imaging, targeting, or controlled release purposes. These attributes make polymer nanoparticles robust and flexible vehicles for hydrophobic drug delivery. Nanoparticle drug delivery systems for cancer treatment is an active field of research both for the medical impact and the complexity of the treatment problems that arise with cancer [2].

Cancer describes a diverse collection of diseases that rely on dysregulated cell signaling pathways to undergo aberrant cell proliferation and survival. While an increasingly wide array of signaling pathways have been found to play roles in the tumorigenesis of various cancers, the phosphoinositide 3-kinase (PI3K) pathway is heavily dysregulated in the majority of cancers, leading to elevated phosphatidylinositol 3,4,5-trisphosphate (PIP_3_) [3,4]. Class I PI3Ks transduce information from receptor tyrosine kinases, G-coupled protein receptors, and Ras-associated GTPases. Due to its role in cell survival, PI3K is a tempting target for therapeutic intervention [3]. The primary targets of intervention within the PI3K kinase family are the class I PI3K heterodimers, which consist of a catalytic subunit and a regulatory subunit. PI3K inhibitors target the catalytic subunit to prevent the kinases from phosphorylating phosphatidylinositol 4,5-bisphosphate (PIP_2_) and converting it into phosphatidylinositol 3,4,5-trisphosphate (PIP_3_) [3]. PIP_3_ interacts with protein kinase B (AKT), leading to a variety of downstream signaling cascades involved in cell survival, motility, proliferation, and metabolism. Inhibiting this pathway halts cell cycle progression and increases apoptosis by disrupting the transduction of critical growth factor signals. PI3K pathway inhibition has also been found to sensitize cancer cells to various chemotherapeutic agents such as doxorubicin, cisplatin, and paclitaxel due to its central role in many critical cellular functions. PI3K inhibitors are either pan-inhibitors that target all of the class I PI3Ks or isoform-specific inhibitors that only interact with one of the catalytic subunits (p110α, p110β, p110γ, or p110δ) (Table 1). The inhibitors listed in Table 1 are all hydrophobic, as described by their water–octanol partition coefficients (log P). Hydrophobic therapeutics have limited bioavailability compared with their hydrophilic counterparts. They exhibit shorter circulation lifetimes since they rapidly sequester in various hydrophobic compartments within the body [5,6].

The first inhibitors that were developed, wortmannin and LY294002, are pan-inhibitors of PI3K. While both wortmannin (1 nM) and LY294002 (1.4 μM) exhibit low half-maximal inhibitory concentrations (IC_50_), they also display poor pharmacokinetics and undesirable toxicity, which are qualities displayed by the pan-inhibitors due to the ubiquitous nature of PI3K in human cells. To date, there are 4 class I PI3K inhibitors that have received FDA approval: alpelisib, idelalisib, copanlisib, and duvelisib [3,4]. Each of these inhibitors is isoform-specific rather than the more potent pan-PI3K inhibitors. Due to PI3K isoform switching, which is commonly observed in cancer, pan-PI3K inhibitors are far more effective at treating tumors than isoform-specific inhibitors, but the undesirable toxicity has presumably prevented any pan-inhibitors from achieving FDA-approval. This characteristic makes pan-PI3K inhibitors such as LY294002 ideal candidates for incorporation into nanoparticle drug delivery systems. Encapsulation will provide an effective increase in soluble drug within the bloodstream and the nanoparticle carrier can provide control over the drug release, reducing system-wide toxicity.

A handful of prior studies exist that examine the feasibility of encapsulating LY294002 within drug delivery vehicles, both nanoscale and microscale. Harfouche and coworkers encapsulated LY294002 at ~0.3 wt% within PLGA nanoparticles (~125 nm) using the single-emulsion solvent evaporation method [9]. They found that B16/F10 cells (mouse melanoma) were susceptible to the nanoparticles and displayed enhanced apoptosis, while MDA-MB-231 cells (human breast adenocarcinoma) only displayed a cytotoxic effect after 72 h at high LY294002 concentrations. To better understand the difference in the LY294002 response between the two cell lines, Harfouche and coworkers studied the internalization of FITC-labeled nanoparticles [9]. MDA-MB-231 cells showed greatly delayed nanoparticle internalization compared with B16/F10 cells, based on a fluorescence assay using LysoTracker Red. The results of these cell studies were replicated in a zebrafish embryo tumor xenograft model. The FITC-labeled nanoparticles were localized to the tumor site and inhibited angiogenesis compared with the vehicle control. Coupled with particle internalization data, Harfouche et al. hypothesized that the therapeutic resistance displayed by the MDA-MB-231 cell link was due to either insufficient internalization of the nanoparticles or the enhanced export of LY294002 out of the cells. The authors’ hypothesized that explanations for their results could explain the observed therapeutic resistance. LY294002 acts on the PI3K pathway within the cell. If LY294002 was rapidly exported from the cell or failed to enter the cell initially, PI3K pathway signaling would not be disrupted.

In a different study, Xu et al. focused on combining LY294002 therapy with DNA demethylation in a human mammary epithelial cell (MCF-10A) cell model [10]. LY294002 was encapsulated (2.1%) within Nile Red labeled PLGA microspheres (1.34 μm) using a microemulsion method. While encapsulated LY294002 did not yield greater cell death compared with the free drug, the PLGA microparticles displayed prolonged LY release over the course of six days. This study did not address the internalization mechanisms, if any, that the microspheres utilized to enter cells.

LY294002 was encapsulated within doxorubicin-conjugated hyperbranched polyacylhydrazone (HPAH-DOX) micelles by Saiyin and coworkers using nanoprecipitation [11]. The HPAH-DOX micelles had a diameter ~40 nm, a positive zeta potential, and LY294002 loading ~5.7 wt%. The micelles exhibited prolonged stability at neutral pH, with only ~28% of encapsulated LY294002 released after 100 h. The HPAH acted as a pH-sensitive trigger for drug release. At pH ~5, nearly 100% of encapsulated LY294002 was released within 40 h. Greater cell death was observed using micelles containing both doxorubicin and LY294002 compared with the physical mixture of HPAH-DOX micelles and unencapsulated LY294002. These results suggest that unencapsulated LY294002 is poorly internalized. This study characterized micelle internalization within CAL-27 and HN-6 tumor cells using fluorescence microscopy by substituting the hydrophobic fluorescent probe, Rhodamine 123 (Rh123), for LY294002 [11]. Rh123 displayed greatly enhanced internalization when it was included within the HPAH-DOX micelles. It was hypothesized that the positive zeta potential of the micelles was responsible for the internalization of the micelles.

Feng et al. investigated nanoliposomes as a delivery vehicle for LY294002 and 5-fluorouracil combination therapy for esophageal squamous cell carcinoma (ESCC) in both in vitro and in vivo studies [12]. Their nanoliposomes were 150 nm in diameter when loaded with the therapeutics at a combined loading level of ~16% *w*/*w*. The nanoliposomes exhibited enhanced cytotoxic activity compared with the physical mixture of unencapsulated LY294002 and 5-fluorouracil, suggesting that the nanocarrier enhanced the treatment. Feng and coworkers investigated nanoliposome uptake using rhodamine B loaded liposomes. They suggested that the fluorescent signal in the cytosol surrounding the nucleus was indicative of endocytosis-mediated cellular uptake [12]. In an in vivo ESCC model, the nanoliposomes significantly reduced tumor growth compared with the combination of both free drugs.

In another study, Cai and coworkers coloaded PLGA nanoparticles (155 nm) with docetaxel (1.7 wt%) and LY294002 (1 wt%) using the single emulsion solvent evaporation technique [13]. The particles exhibited a rapid release (55%) of LY294002 in vivo within the first 24 h, followed by slow steady release for the remainder of the release study (5 days). Nanoparticle biodistribution by IVIS following tail vein injection was evaluated using the fluorescent label DiR. Free DiR was not observed at the tumor site, but DiR-loaded PLGA nanoparticles localized to the tumor site within the first 2 h and were retained for up to 120 h. In a tumor xenograft mouse model using MKN45 cells, docetaxel (10 mg/kg) alone, docetaxel and free LY (15 mg/kg), or PLGA NPs containing both docetaxel and LY were administered by tail vein injection. After 30 days, the drug-loaded PLGA NP group exhibited tumors with significantly lower weights compared with all other groups.

This prior work shows the advantages of encapsulating LY294002 within a drug delivery vehicle, but, for clinical use, additional work is still needed to increase the LY294002 loading capacity and to control the size of nanocarriers. Several different methods exist to synthesize polymer nanoparticles: single emulsion, double emulsion, coacervation, nanoprecipitation, and flash nanoprecipitation (FNP) [14,15,16,17]. Of all these methods, FNP offers the best combination of high drug loading, particle size control, and the potential for post-fabrication functionalization for targeted delivery. FNP involves the rapid and turbulent mixing of miscible solvent streams, where one or more of the streams serves as an anti-solvent for the solutes in a different stream. The solutes rapidly precipitate and form nanoparticles due to the high solute supersaturation within the mixed solvent system. FNP produces polymer nanoparticles with well-defined size distributions and high drug loadings by rapidly creating homogeneous supersaturated mixtures. Supersaturations greater than 1000 are possible with FNP [18]. FNP is especially well suited for the encapsulation of highly hydrophobic (log P > 6) therapeutics [18,19], as well as hydrophobic dyes that are efficiently incorporated in the particle cores [20], for tracking particle transport. The polymer nanoparticles detailed in the present work were found to have higher LY wt% loadings relative to the values reported in the literature. The higher levels of encapsulated LY294002 may help overcome the LY294002 resistance described in the work by Harfouche [9].

Encapsulating mildly hydrophobic therapeutics, with partition coefficients (logP) between 0 and 3.5, remains a challenge when employing FNP due to the difficulty in generating sufficiently high supersaturations of the therapeutics during the mixing process. Insufficient supersaturation leads to limited and inefficient sequestration of the therapeutics out of the solvent phase into the growing nanoparticle phase. A method that would increase the supersaturation of this class of therapeutics during flash nanoprecipitations would allow for the encapsulation of a much wider array of therapeutic agents. Hydrophobic ion pairing is a technique for increasing the supersaturation of a therapeutic during flash nanoprecipitation by reducing its solubility in the anti-solvent. The technique involves mixing the therapeutic with a hydrophobic counter ion to form a salt that is much more hydrophobic than the original free therapeutic. For efficient ion pairing, the acidic IP should have a pKa ~2 pH units smaller than the free base API [21]. Drug release rates can be controlled by adjusting the IP:API molar ratio used during nanoparticle synthesis. This technique [22] has been shown to produce high loadings of siRNA [23], polymyxin B [24], and the experimental single-dose malaria therapeutic OZ439 [25].

Using hydrophobic ion pairing (IP) agents (hexadecylphosphonic acid (HDPA) and palmitic acid (PA)), we examined the effectiveness of this technique on increasing the loading capacity of the model mildly hydrophobic compound LY294002, a PI3K pathway inhibitor, in PEG-PDLLA nanoparticles. We hypothesized that the use of a hydrophobic ion pairing agent would produce nanoparticles with higher LY294002 loading levels than when LY294002 is used alone. Additionally, we suspected HDPA would result in higher LY294002 encapsulation than PA and in particles with a more well-defined size distribution since HDPA has a pKa value several pH units less than PA. The size of nanoparticle drug formulations affects their circulation time following intravenous injection and determines the bioavailability of the encapsulated therapeutic; thus, we explored how the choice of ion pairing agent affects nanoparticle size. Using a novel in vitro release protocol developed by Shen and coworkers, we were able to examine the release profile of LY294002 in the presence of hydrophobic sink conditions [26]. All particles in this work also included a hydrophobic dye—6,13-Bis(triisopropylsilylethynyl) pentacene (TIPS)—which previous work had shown could be incorporated into particles with PDLLA cores at ~100% loading efficiency [27] and could be used to track particle transport, particularly particle uptake by cells such as macrophages [28]. This work will increase our understanding about how to engineer polymer nanoparticles with high loading capacities of LY294002 with flash nanoprecipitation and hydrophobic ion pairing. In view of these objectives and the prior literature, the design constraints for a NP-based delivery system are summarized in Table 2.

## 2. Materials and Methods

### 2.1. Materials

The LY294002 (Cat. No. L-7962, 307.35 g/mol) and LY294002 HCl (Cat. No. L-7988, 343.80 g/mol) were purchased from LC Laboratories, Woburn, MA, USA. Tetrahydrofuran (THF) (CAS 109-99-9; SKU 401757), palmitic acid (PA) (CAS 57-10-3, 256.42 g/mol), hexadecylphosphonic acid (HDPA) (CAS 4721-17-9, 306.42 g/mol), and 6,13-Bis(triisopropylsilylethynyl) pentacene (TIPS) (CAS 373596-08-8) were purchased from Sigma-Aldrich, St. Louis, MO, USA. Methoxy-poly(ethylene oxide)-b-poly(D,L lactic acid) (PEO(5k)-b-PDLLA(8.9k); DP_PEO_ = 114, DP_PDLLA_ = 123) was synthesized using methods described previously [29] and stored as a dry powder at room temperature in a desiccator. Milli-Q water (~18 MΩ-cm) was produced from a Millipore Synergy Ultrapure Water system.

### 2.2. Nanoparticle Formation

Nanoparticles (NPs) comprising PEO-b-PDLLA copolymer with LY were made using FNP (Figure 1) performed with a confined impingement jet mixer with dilution (CIJ-D) [30]. An exemplary case is described below for NPs containing LY and TIPS. Solutions of PEO-b-PDLLA diblock copolymer were prepared in THF at concentrations of 50 mg/mL. The solutions were sonicated for 30 min in an ultrasonic bath (110 W at 40 kHz) at ~25 °C. The solutions were diluted into scintillation vials containing LY and either HDPA or PA and THF. HDPA and PA were omitted from the formulation for the LY-only control nanoparticles. A solution of the fluorescent dye TIPS pentacene dissolved in THF was prepared to a final solution concentration of 2 mg/mL. The vial containing the TIPS solution was wrapped in foil to prevent degradation. Aliquots of the TIPS solution were diluted into the samples to achieve a final TIPS concentration of ~0.33 mg/mL. The final mixture of TIPS and the copolymer was blue without any trace of cloudiness, indicating complete dissolution of the solutes.

The resulting solutions in THF were sonicated for 30 min. When the solutions contained TIPS, the ultrasonic bath was covered to prevent photodegradation. After sonication, the solutions were filtered through a 0.2 μm syringe filter (nylon, Fisher Scientific, Waltham, MA, USA, Cat. No. 09-719-006). The filtered solutions were then loaded into 2.5 mL Hamilton gastight syringes and all air bubbles were removed. An identical volume of Milli-Q water was added to a second 2.5 mL gastight syringe. The organic solution was injected into the CIJ-D (Figure 1) against an equivalent volume of antisolvent (Milli-Q water). The mixer effluent was caught in a 40-mL glass vial containing sufficient Milli-Q water to achieve a final solvent volume ratio of 1:10 THF:water. Following the mixing process, the samples were dialyzed against ~1000× greater volume of water for 24 h, during which the water was changed 4 times. After dialysis, the samples were transferred to centrifuge tubes and frozen in either a bath of acetone and dry ice or a freezer at −70 °C. The frozen samples were placed on a lyophilizer (−50 °C, 0.08 mbar) for at least 3 days and then the lyophilized samples were transferred to 20 mL scintillation vials and stored at 4–8 °C.

Based on our prior work, dimethylformamide (DMF) is also a suitable solvent for both PEO-b-PDLLA and TIPS. DMF is substantially more polar than THF, with a dielectric constant ~5× greater than THF. Solvent polarity could affect the solubility of both LY and the ion pairing agents during nanoparticle synthesis. We also tested the effect of using DMF as the organic stream solvent on the characteristics of the resulting nanoparticle populations. In these cases, all sonication prior to FNP was performed at 45 °C to facilitate the dissolution of the hydrophobic ion pairing agents in DMF. The organic solutions were injected into the CIJ-D about 30 s after being removed from the heated ultrasonic bath.

All the nanoparticles in this study were made using a PEO-b-PDLLA diblock copolymer concentration of 32 mg/mL. For nanoparticles (NPs) containing only TIPS dye at a target loading of 1 weight%, the final solution comprising the TIPS and the diblock used in FNP had a TIPS concentration of ~0.33 mg/mL. For nanoparticles containing only LY at a target loading of 20 wt%, the final solution had an LY concentration of ~8 mg/mL. For nanoparticles containing LY and PA at an LY target loading of 20 wt% and an LY:PA molar ratio of 1:1, the final solution had an LY concentration of ~10 mg/mL and a PA concentration of ~8.3 mg/mL. The molar ratio of LY:PA was also studied at 2:1, 3:1, and 4:1. For nanoparticles containing LY at a target loading of 20 wt%, the final solution had an LY concentration of ~10.5 mg/mL and an HDPA concentration of ~10.5 mg/mL for an LY:HDPA molar ratio of 1:1. 

### 2.3. Dynamic Light Scattering (DLS)

DLS was performed on all samples before and after dialysis as well as on samples that were dispersed following freeze drying. For analyses before dialysis, typically 100 μL of each sample suspension was added to 3 mL of Milli-Q water and the resulting suspension was then sonicated for 30 min. Approximately 1 mL of the suspension was pipetted into a disposable cuvette and DLS measurements were performed at 25 °C using a Zetasizer NanoZS (Zetasizer Software v7.12, Malvern Panalytical Inc., Westborough, MA, USA). For analyses after dialysis, 100 μL of each sample suspension was added to 4 mL of Milli-Q water and the resulting suspension was then sonicated for 30 min. DLS measurements were performed using the same procedure as detailed for the before dialysis measurements. For analyses after freeze-drying, 5 mL of Milli-Q water was added to the centrifuge tubes used during the freeze-drying process to collect the dry samples. The centrifuge tubes were vortexed for 10 s and then 1 mL of each suspension was diluted with 3 mL of Milli-Q water and sonicated for 30 min. 

### 2.4. Zeta Potential

The zeta potential of each sample was measured after redispersion following freeze-drying. The measurements were performed on the same samples used for DLS measurements. Typically, 1 mL of a sample suspension was added to a disposable polystyrene folded capillary cell. The zeta potential measurements were performed using a Zetasizer Nano ZS (Zetasizer Software v7.12) at 25 °C.

### 2.5. UV/Vis Spectrophotometry

The drug loading of the nanoparticle batches was determined with UV/Vis spectrophotometry (ThermoFisher Scientific/EV3-154601, Waltham, MA, USA). Calibration curves were created for both TIPS and LY, where the absorbances were measured at wavelengths of 641 nm and 299 nm, respectively. Freeze-dried nanoparticles were transferred to 20 mL scintillation vials and dissolved in THF to a concentration of 2 mg/mL. These samples were diluted as needed to measure absorbance peaks within the bounds of the calibration curves.

The primary LY absorbance peak is found at a wavelength of 299 nm (Appendix A). For each replicate, the dilution with the highest concentration that yielded a curve within the linear absorbance regime at 299 nm (absorbance < 1) was selected to determine the amount of released LY. The shape of the absorbance spectra and the location of the peaks did not change for nanoparticles created with in situ ion pairing when compared with the drug-only control nanoparticles. This suggests that the presence of an ion pairing agent does not influence the measured absorbance of LY during the release study. 

All absorbance measurements were normalized against the THF background absorbance. We used a scan rate of 120 nm/min over a wavelength range of 200–800 nm. The measured absorbance values were compared with the calibration curves to determine the final solute concentration. The wt% loading of the nanoparticle sample was then calculated according to:(1)$$Drug\;Loading=\left(\frac{Mass\;of\;Drug\;in\;the\;Nanoparticles}{Mass\;of\;the\;Nanoparticles}\right)*100$$

The encapsulation efficiency of the nanoparticle sample was calculated according to:(2)$$Encapsulation\;Efficiency=\left(\frac{Mass\;of\;Drug\;in\;the\;Nanoparticles}{Mass\;of\;Drug\;in\;the\;Organic\;Stream}\right)*100$$

For samples that contained both LY and TIPS dye, determining the LY loading was complicated by the absorbance of TIPS at 299 nm. The absorbance of these samples was measured at two wavelengths and then the concentrations of both LY and TIPS were calculated simultaneously using Vierordt’s method:(3)$$C_{LY}=\frac{A_{641}a_{TIPS,299}-A_{299}a_{TIPS,641}}{a_{LY,641}a_{TIPS,299}-a_{LY,299}a_{TIPS,641}}$$
(4)$$C_{TIPS}=\frac{A_{299}a_{LY,641}-A_{641}a_{LY,299}}{a_{LY,641}a_{TIPS,299}-a_{LY,299}a_{TIPS,641}}$$
where C_X_ is the concentration of species X, A_299_ and A_641_ are the measured absorbances at 299 nm and 641 nm, and a_X,299_ and a_X,641_ are the absorptivity values of species X at wavelengths of 299 nm and 641 nm, respectively.

If any signs of scattering were observed in an absorbance measurement, a scattering correction was applied to the dataset before using Vierordt’s method. Correcting for scattering is critical for accurately determining the concentrations of chemical species in our samples. A section of the absorbance spectrum was selected such that the only measured absorbance in that region comes from scattering. Using the measured absorbance values from that region, we performed a least-squares fit on the transformed data.
(5)$$A=a\lambda^{n}$$
(6)$$\log\left(A\right)=\log\left(a\right)+nlog\left(\lambda \right)$$

The measured absorbance, A, and the corresponding wavelength, λ, were used and we solved for the constants n and a. Once the values of n and a were known, we calculated the amount of scattering at the wavelengths of interest (299 nm and 641 nm) and normalized our measured absorbance values accordingly.

### 2.6. LY294002 Release

We employed a custom diffusion cell to study the release of LY from the polymeric nanoparticles in the presence of a hydrophobic sink to account for the release of the hydrophobic therapeutic more accurately within the body. This diffusion cell was based on a design employed in an earlier study on the encapsulation and release of curcumin [25]. A nanoparticle suspension was prepared at 5 mg/mL by adding 1× PBS to freeze-dried nanoparticles and sonicating the suspension for 30 min at room temperature. The suspension was added (3 mL) to the diffusion cell. A dialysis membrane (12–14 kDa MWCO) was placed over the suspension and additional caution was taken to prevent any air bubbles from being trapped between the nanoparticle suspension and the dialysis membrane. The membrane was secured using a PTFE o-ring. We utilized a horse-pinch clamp to secure both halves of the diffusion cell together. An additional 2 mL of 1× PBS was added to the diffusion cell above the dialysis membrane. Methyl tert-butyl ether (MTBE) (2 mL), which is immiscible with and less dense than water, was carefully added to the diffusion cell to create an organic phase sitting directly on top of the 1x PBS without disturbing the 1× PBS phase. The MTBE phase served as a hydrophobic sink for LY during the release study. At each timepoint, 500 μL aliquots were taken from the MTBE phase and replaced with fresh MTBE. The aliquots were analyzed using a UV/Vis spectrophotometer at 299 nm to determine the LY content of each sample.

### 2.7. Statistical Analysis

All statistical analysis was performed using either Excel or JMP Pro 13 software. Unless otherwise specified, all statistical comparisons were performed with α-values of 0.05.

## 3. Results

### 3.1. LY294002 Incorporation in PEO-PDLLA Nanoparticles without Ion Pairing

Given the hydrophobic nature of LY294002 (Log P = 3.34, pKa = 3.47), we sought to determine whether FNP could create polymer nanoparticles with LY294002 trapped within their hydrophobic cores. The resulting nanoparticles were analyzed using dynamic light scattering (DLS), zeta potential measurements, and UV/Vis spectrophotometry (Table 3). The LY-only nanoparticles made with DMF were smaller than those made with THF (Figure 2), suggesting that PDLLA is less soluble in the DMF:water mixture, which would result in more rapid precipitation of the PDLLA into nucleating LY particles. The necessity to synthesize the nanoparticles at 45 °C in the DMF cases convolutes the effects of temperature and solvent, so it is difficult to determine how mixing temperature influenced these results. The polymer nanoparticles had an LY294002 loading of 3.78 wt% and 0.53 wt% when made with THF at 25 °C and DMF at 45 °C, respectively.

The low encapsulation efficiency suggests two possibilities: (1) LY294002 is not sufficiently hydrophobic to sequester into the cores of the growing polymer nanoparticles or (2) LY294002 is being released from the polymer nanoparticles during the 24 h dialysis step. LY294002 is insufficiently hydrophobic to efficiently partition into the growing nanoparticle phase after nanoparticle nucleation but could adsorb to the surface of the nanoparticles. Any adsorbed LY would quickly desorb and diffuse away from the nanoparticles during the extended dialysis period. To address the undesirable loading levels of LY, we investigated how the use of ion pairing (IP) agents during flash nanoprecipitation could improve the incorporation of LY in the nanoparticles.

### 3.2. Ion Pairing Agents Increase the LY294002 Loading of PEO-PDLLA Nanoparticles

To explore the effect of the solubility of LY294002 on drug loading, we examined the effect of creating a hydrophobic salt of LY294002 and a hydrophobic acid. Prior work in the literature [21,24,31] suggests that forming a hydrophobic salt of LY and a hydrophobic acid during FNP will yield nanoparticles with higher drug loading levels than those prepared without an ion pairing agent. A hydrophobic drug salt of LY will be less soluble than free LY under the mixed water–organic solvent conditions during nanoparticle synthesis, thus promoting the precipitation of the drug salt into nanoparticle nuclei. We tested two different hydrophobic acids or ion pairing (IP) agents: palmitic acid and hexadecylphosphonic acid. Palmitic acid is a weak acid with a pKa of 4.95, while hexadecylphosphonic acid has a pKa of 1.81. Nanoparticles formed with palmitic acid and LY in THF at 25 °C displayed significant aggregation, whereas PA:LY nanoparticles synthesized in DMF at 45 °C displayed a bimodal size distribution. We suspect the distribution consists of aggregated particles (382 nm) containing palmitic acid and LY mixed with small PEO-PDLLA particles (64 nm). We hypothesize that the LY is primarily contained in the aggregates made with THF rather than the small nanoparticles. The final LY loading was 6.8 wt% and 4.1 wt% for palmitic acid containing nanoparticles made in THF and DMF, respectively. The addition of hexadecylphosphonic acid at a 1:1 molar ratio with LY294002 to THF during FNP yielded polymer nanoparticles with hydrodynamic diameters of ~201 nm and zeta potentials of −40 mV. The LY294002 loading and encapsulation efficiency were 11.1 wt% and 55.3%, respectively. When the organic solvent was DMF, the resulting nanoparticles had hydrodynamic diameters of 213 nm and zeta potentials of −46 mV. The LY294002 loading capacity and encapsulation efficiency were 7.85 wt% and 39.3%, respectively.

We hypothesized that HDPA would lead to higher loadings of LY294002 than PA due to more efficient proton transfer between the acid and LY resulting from the lower pKa value [21,24,31]. The addition of either hydrophobic ion pairing agent significantly increased the LY loading, regardless of the choice of organic solvent used in the organic stream (Figure 3). The choice of organic solvent to use for the organic stream greatly impacted the effects of the hydrophobic ion pairing agents on the encapsulation of LY294002. HDPA significantly increased LY294002 encapsulation compared with PA for nanoparticles fabricated using DMF. No significant difference in LY294002 encapsulation wt% was detected between nanoparticles made with HDPA or PA when THF was used as the organic stream solvent. We hypothesize that the ion pairing agents did not incorporate homogenously when THF was used as the organic solvent, which resulted in the observed LY loading levels. Immediately following FNP with PA, macroscopic precipitation was visible in the mixer effluent. The high levels of LY present in this case may be due to the formation of a hydrophobic PA phase, within which unencapsulated LY was sequestered. The measured LY294002 loading would represent the LY294002 encapsulated within nanoparticles as well as any LY294002 that was sequestered following FNP. One possible explanation for this behavior is crystallization of the ion pairing agents’ carbon chains during FNP. FNP using either ion pairing agent in THF produced polydisperse heavily aggregated particle populations (Table 4), which suggests that the ion pairing agents were incompletely dissolved in THF due to poor solubility. A bimodal size distribution was observed when palmitic acid was used as the ion pairing agent in DMF. In contrast, FNP using DMF produced stable nanoparticle populations with well-defined sizes. We hypothesize that additional heating may alleviate the formation of aggregates by ensuring that the hydrocarbon chains of the ion pairing agents do not crystalize during flash nanoprecipitation. For the THF case there is no noticeable change in PDI between the control and the HDPA cases. In the DMF case, however, there is a slight reduction in the polydispersity in the HDPA cases which we attribute to the difference in polarity of DMF compared with THF. The higher dielectric constant of DMF may make it a better solvent for LY than THF.

### 3.3. Release of LY from PEO-PDLLA-Salt Former Nanoparticles 

We investigated the LY release kinetics from the PEO-PDLLA nanoparticles over 1 week. All three nanoparticle cases displayed prolonged LY release over the entire week (Figure 4). We approximated an infinite hydrophobic sink, analogous to that found within the human body, using a solvent system of 1× PBS and MTBE in the modified diffusion cell based upon a previous study [25]. Nanoparticles formed with either PA or HDPA released a greater percentage of their total LY load compared with the LY-only control. Nanoparticles formed with HDPA displayed almost a linear drug release over the course of the study (7 days). The measured drug release kinetics are consistent with studies on FNP-synthesized polymer nanoparticles from the literature [22,24,32,33,34,35]. Generally, drug release from FNP-synthesized nanoparticles exhibits three phases: (i) initial burst-release phase, (ii) slow-release phase, and (iii) plateau phase. The nanoparticles exhibited steady release over the course of the study and did not display an initial burst release of LY. The addition of an ion pairing agent shifted the observed LY release kinetics towards approximately linear (first order) behavior. We suspect that the inclusion of hydrophobic ion pairing agents extended the LY release from the nanoparticles. This would be consistent with the findings of Ristroph et al. [22,24]. The steady and prolonged drug release exhibited by these nanoparticles is a beneficial feature for drug delivery applications. Additional work is needed to make any assertions regarding the in vivo circulation time that these polymer nanoparticles will display following IV injection. D’Addio and coworkers found that nanoparticle size, diblock copolymer composition, and polymer block molecular weights influenced the circulation time of drug-loaded polymer nanoparticles synthesized with FNP [29]. In the future, in vivo studies should be performed to determine how LY-loaded polymer nanoparticles behave in circulation following IV administration and to characterize the length of time the nanoparticles remain in circulation.

## 4. Conclusions

LY294002 is too hydrophobic to use clinically by itself and insufficiently hydrophobic to achieve adequate loading in polymer nanoparticles fabricated using flash nanoprecipitation alone. By adding a hydrophobic ion pairing agent, hexadecylphosphonic acid, we were able to achieve drug loadings of LY294002 in polymer nanoparticles that are higher than have been reported previously in the literature [11] and are likely clinically relevant. The nanoparticles displayed release kinetics that approached linear kinetics, with the majority of the encapsulated API releasing steadily over the course of a week. While further optimization of the flash nanoprecipitation mixing parameters is needed to achieve nanoparticles with diameters below 100 nm, we have demonstrated that flash nanoprecipitation using hexadecylphosphonic acid as an ion pairing agent can produce polymer nanoparticles with clinically relevant levels of hydrophobic therapeutics.

## Figures and Tables

**Figure 1 pharmaceutics-15-01157-f001:**
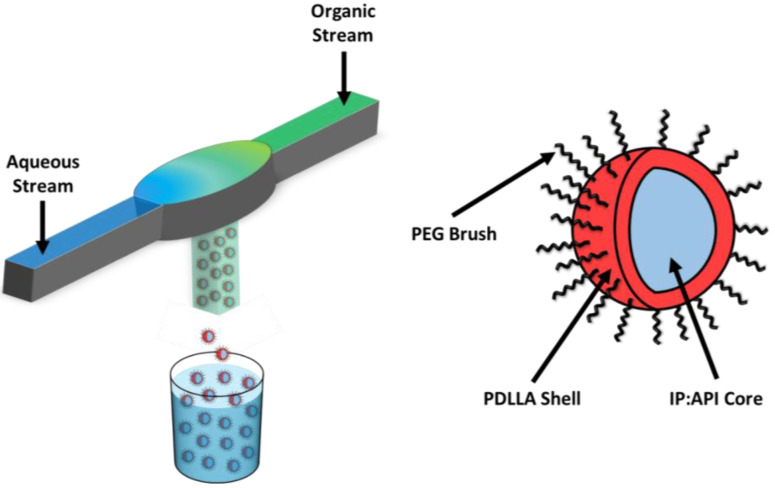
Synthesis of LY294002-loaded PDLLA-PEG nanoparticles (GEM-Chit NPs) with flash nanoprecipitation. Two streams undergo turbulent mixing within the mixing chamber, forming the core of the nanoparticles. The nanoparticles are predicted to exhibit a core–shell structure with the IP:LY salt forming the nanoparticle core and the PDLLA-PEG forming a stabilizing corona.

**Figure 2 pharmaceutics-15-01157-f002:**
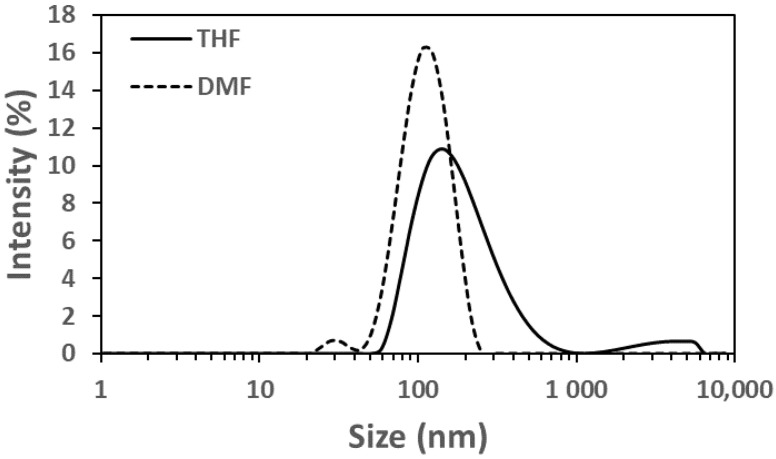
Intensity distributions of LY-containing nanoparticles made without ionic pairing agents. The solid line represents the nanoparticles made using THF at 25 °C and the dotted line represents those made with DMF at 45 °C.

**Figure 3 pharmaceutics-15-01157-f003:**
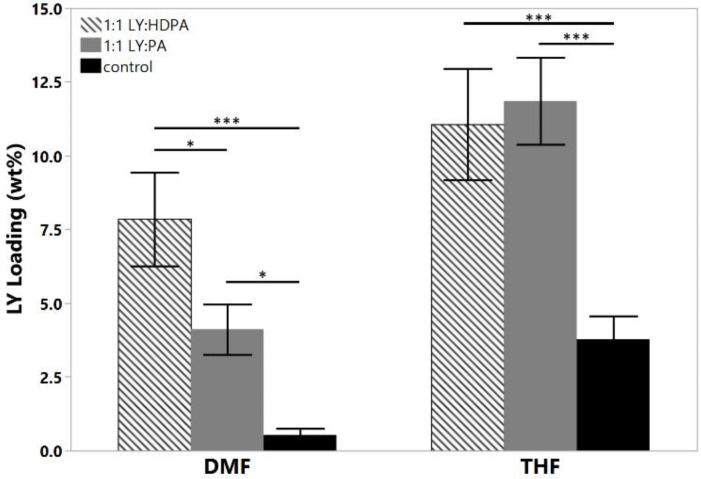
Effect of hydrophobic ion pairing on LY294002 encapsulation. Both HDPA and PA significantly increased the final LY loading compared with the LY only controls in both solvents. The target LY loading was 20 wt% in each case; (* *p* < 0.05; *** *p* < 0.005).

**Figure 4 pharmaceutics-15-01157-f004:**
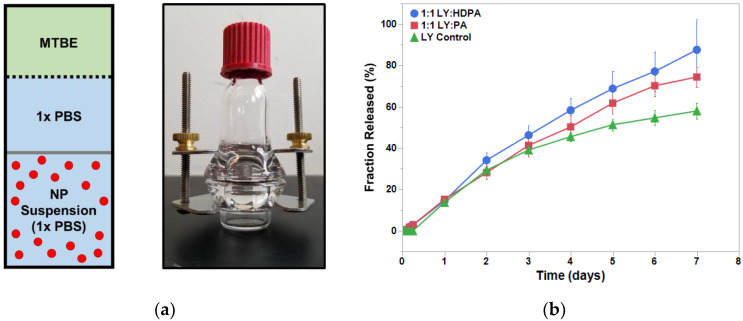
LY294002 release in the presence of IPs. (**a**) Schematic and an image of the diffusion cell. The dialysis membrane (12–14 kDa MWCO) is represented by the gray bar and the dashed line represents the MTBE-PBS interface. (**b**) Results of the 1-week release study for the LY-only NPs (triangles), LY:PA NPs (squares), and LY:HDPA NPs (circles). MTBE— methyl tert-butyl ether.

**Table 1 pharmaceutics-15-01157-t001:** Drug delivery characteristics and FDA-approval status of PI3K pathway inhibitors.

Inhibitor	Target	IC_50_ ^6^	Log P ^5^	FDA Approval	Administration Notes
Alpelisib	p110α	5 nM ^1^	3.81	Approved	Oral dosage in combination with fulvestrant ^2^
Idelalisib	p110δ	2.5 nM ^1^	3.68	Approved	Oral dosage ^3^
Copanlisib	p110α/δ	0.5 nM, 0.7 nM [7]	0.5	Approved	1-h intravenous infusion ^4^
Duvelisib	p110γ/δ	27 nM, 2.5 nM [8]	4.56	Approved	Oral dosage
Wortmannin	Pan-PI3K	1 nM [3]	1.84	---	
LY294002	Pan-PI3K	1.4 μM [3]	3.64	---	

^1^ Selleck Chemicals; ^2^ Novartis product page; ^3^ Gilead prescription information document; ^4^ Bayer prescription information document; ^5^ Octanol/water partition coefficients calculated using Molinspiration; ^6^ Drug concentration that reduces the target kinase activity to 50% of the maximal activity.

**Table 2 pharmaceutics-15-01157-t002:** Nanocarrier design criteria for delivery of LY294002.

Criterion	Discussion
Cell type(s) affected by drug (LY294002)	All human cells
Mode of administration	Intravenous, intraperitoneal
Drug dosage	IC_50_: 1.4 µM Mouse study: 20 mg/kg Human: unknown
Location of therapeutic effect	Intracellular
Cellular membrane transport limitation?	No
Intracellular processing of NPs	Lysosomal release
Polymer carrier	PDLLA-PEG
Drug loading range	10 (mg NP)/mL IP injection at 50 wt% LY loading to reach 20 mg/kg dosage
Drug Log P at drug release pH	3.64 (Log P)
Drug pKa	3.47
Hydrophobic ion pairing agent	HDPA (pKa: 1.81) PA (pKa: 4.95)
Solvent(s) suitable for drug dissolution in FNP process	Tetrahydrofuran, dimethylformamide
Drug release kinetics	Burst release phase followed by slow, sustained release
Importance of triggered release	pH-triggered release below pH 6.5

**Table 3 pharmaceutics-15-01157-t003:** Properties of nanoparticle components.

Chemical Name	Structure	pKa	Log P
PDLLA-PEG	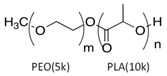	--	--
LY294002 (LY)	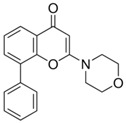	3.47	3.34
Palmitic Acid (PA)	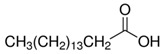	4.95	7.60
Hexadecylphosphonic Acid (HDPA)	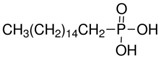	1.81	5.13
TIPS Pentacene	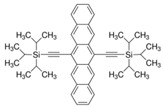	--	9.28

**Table 4 pharmaceutics-15-01157-t004:** DLS and UV/Vis spectroscopy characterization of PDLLA-PEG nanoparticles.

Case	Intensity Average Diameter (nm)		Polydispersity Index		Zeta Potential (mV)		LY294002 Loading (wt%) ^2^	
	THF	DMF	THF	DMF	THF	DMF	THF	DMF
LY-only Control	260 ± 130	128 ± 30	0.35 ± 0.10	0.34 ± 0.13	−33 ± 3	−29 ± 2	3.8 ± 0.7	0.5 ± 0.2
1:1 LY:PA ^1^	--	382 ± 78 64 ± 8	--	--	−35 ± 2	−41 ± 2	12 ± 1	4.1 ± 0.9
1:1 LY:HDPA	335 ± 34	213 ± 16	0.32 ± 0.04	0.23 ± 0.04	−38 ± 3	−46 ± 2	11 ± 2	7.9 ± 2

^1^ Too polydisperse for reliable DLS measurement. ^2^ Target loading was 20 wt% LY in all cases.

## Data Availability

The datasets analyzed in this work are housed locally and are available upon request.

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
