# Peer review of "Encapsulation of PI3K Inhibitor LY294002 within Polymer Nanoparticles Using Ion Pairing Flash Nanoprecipitation"

_pharmaceutics, 2023, doi:10.3390/pharmaceutics15041157_

Round 1

Reviewer 1 Report

Reviewing Report

The current research work entitled "Encapsulation of PI3K inhibitor, LY294002, within Polymer Na- 2noparticles using Ion Pairing Flash Nanoprecipitation" consider an novel method for the preparation of polymeric nanoparticles loaded with hydrophobic drugs containing ionizable groups. The work investigated several parameters that can affect the efficiency of incorporation. The manuscript should be revised thoroughly with respect to English language and the author should address the following;

1- Although there is a significant difference in loading efficiency between the used ion paring and solvent, the loading efficiency of the tested API is still low where about 12% of added API were encapsulated. Could the authors explain?

2- What about PDI of the prepared NPs, why the author did not present and discuss it.

3- page 11, line 447 what is MTBE refer to

4- Further investigation such FTR, NMR and HPLC are need to omit the chemical interaction between LY294002 and used additives, specially, ion pairing agents.

5- Imaging of the produced NPs by TEM or SEM is strongly required.

6- The explanation of the effect of the used solvent, THF and DMF should be deeper

7- The kinetic of drug release form different NPs should be done.

8- A preliminary water-octanol partition coefficients (log P) experiment in the presence of ion pairing agents will be helpful and gives an idea about the effect of ion pairing on changing the affinity of API to octanol and water. The results of this experiment will be strongly indicative.

Reviewer 2 Report

MANUSCRIPT – Pharmaceutics-2168695

In this article, Encapsulation of PI3K inhibitor, LY294002, within Polymer Nanoparticles using Ion Pairing Flash Nanoprecipitation the authors have proposed a methodology to prepare polymeric nanoformulations using flash precipitation along with the ion pairing agents for encapsulating LY294002. This investigation studied the effect of 2 hydrophobic IPs, which have a direct role in drug loading efficiency, their size, and the solvent's impact on producing highly colloidal and well-defined formulations. This design methodology is exciting and can be implemented in the pharmaceutical industry.  

 Significant comments that should be addressed

 Title

Suggestion-better to replace a comma with a hyphen in the title. For example, PI3K inhibitor- LY294002

Introduction

Line 40 – I would recommend including this reference for nanoparticles-based DDSs for cancer (https://doi.org/10.3390/pharmaceutics13111803).

In general, the introduction section is too extensive; authors would consider minimizing it if possible.

Materials and Methods

Section 2.2, please provide a schematic illustration that explains the whole process, as it is prolix.

Section 2.4-change capillary to folded capillary cell

Results

Section 1.2-Explain briefly the advantage of adding 2 ion pairing agents.

In most cases, the degree symbol is missing for °C.

Section 1.3-why the drug release studies been restricted to 7 days?

Recommendation-Please validate your drug release data with some of the kinetic models and represent the graph or numerical values.

Why is the drug release study restricted to only 1 pH? Usually the release studies will be carried out at different pH.

Conclusion

The authors have mentioned that the size will be optimized to below 100 nm. Can you explain how briefly?.

Reviewer 3 Report

In this article, the authors explored the encapsulation of the PI3K inhibitor(LY294002) in poly(ethylene glycol)-b-poly(D,L lactic acid) nanoparticles using flash nanoprecipitation. The engineered polymer nanoparticles demonstrated high loading capacity of LY294002. In addition, the authors also investigated the effects of hydrophobic ion pairing agents on the size of nanoparticles and the loading of LY294002. The results show that nanoparticles with a LY 294002 load of 7.9% could be prepared by selecting organic solvent DMF and using hydrophobic ion pairing agent HDPA. The wrapped active pharmaceutical ingredients could be released stably for a week. I think this study is very interesting and the results are reliable. I would like to recommend it to be published after minor revision.

Here are several questions and/or suggestions the authors may consider.

1.    In line 368, 413, 446, the figure numbers seem to be wrong. It seems the figure 1 was not mentioned in the article.

2.    In line 232, it is mentioned that “an exemplary case is described below for NPs containing LY and TIPS”, However, I could not find a clear indication from the text that in which step LY was added in. Please improve the clarity of the description of the preparation method.

3.    In line 454, it is mentioned that “the release curves of the nanoparticle formulations containing hydrophobic ion pairing agents enter the slow-release phase proceeding the plateau between day 6 and day 7”. However, in Figure 4 (b), the slope of the release curve (blue and red curves) between day 6 and day 7 seems to be similar to that of first several days. Therefore, please explain how this conclusion was reached.

Round 2

Reviewer 1 Report

no comments